# Robotic Grasp Detection Network Based on Improved Deformable Convolution and Spatial Feature Center Mechanism

**DOI:** 10.3390/biomimetics8050403

**Published:** 2023-09-01

**Authors:** Miao Zou, Xi Li, Quan Yuan, Tao Xiong, Yaozong Zhang, Jingwei Han, Zhenhua Xiao

**Affiliations:** 1School of Electrical and Information Engineering, Wuhan Institute of Technology, Wuhan 430205, China; zoumiao@stu.wit.edu.cn (M.Z.); hanjingwei@stu.wit.edu.cn (J.H.); 2College of Information and Artificial Intelligence, Nanchang Institute of Science and Technology, Nanchang 330108, China; xiakyxiong@163.com (T.X.); zhangyaozong@wit.edu.cn (Y.Z.); xiaozh@hbc.edu.cn (Z.X.)

**Keywords:** grasp detection, deformable convolution, spatial feature center mechanism, robotic arm

## Abstract

In this article, we propose an effective grasp detection network based on an improved deformable convolution and spatial feature center mechanism (DCSFC-Grasp) to precisely grasp unidentified objects. DCSFC-Grasp includes three key procedures as follows. First, improved deformable convolution is introduced to adaptively adjust receptive fields for multiscale feature information extraction. Then, an efficient spatial feature center (SFC) layer is explored to capture the global remote dependencies through a lightweight multilayer perceptron (MLP) architecture. Furthermore, a learnable feature center (LFC) mechanism is reported to gather local regional features and preserve the local corner region. Finally, a lightweight CARAFE operator is developed to upsample the features. Experimental results show that DCSFC-Grasp achieves a high accuracy (99.3% and 96.1% for the Cornell and Jacquard grasp datasets, respectively) and even outperforms the existing state-of-the-art grasp detection models. The results of real-world experiments on the six-DoF Realman RM65 robotic arm further demonstrate that our DCSFC-Grasp is effective and robust for the grasping of unknown targets.

## 1. Introduction

With the rapid advancement of artificial intelligence technology, the application of robots has gradually shifted from the traditional industrial field to unstructured environment fields, such as home services, warehousing, and logistics, expanding the range of application scenarios for intelligent grasp detection in robotic arms. Grasping is a basic robot skill. Achieving robot grasping intelligence can significantly improve production efficiency and human–computer interactions. Compared with traditional industrial robots that repeatedly grasp specific known objects in fixed working areas, robot grasping in unstructured environments encounters many practical challenges, such as changes in lighting and scenery, unknown operation examples, and diverse placement positions. Visual perception is an important process for intelligent robots to understand the real world. Enhancing visual perception can help robots overcome grasping challenges in unstructured environments and further improve their adaptability to grasp new objects in new environments.

To determine precise grasp positions, the robot should focus not only on local geometric features but also on the entire visual exterior of the target. Particularly in irregular and chaotic conditions, adapting to changes in appearance, location, and spatial relationships with adjacent objects is vital for grasp detection performance. In some previous grasp detectors [1,2], most convolutional neural networks had to repeatedly stack convolutional layers to maintain a large receptor field, which diminished the spatial resolution ratio and unavoidably led to the disappearance of global particulars and performance degradation. In recent years, transformers [3,4,5] have been successfully applied as new methods for computer vision. The transformer attention mechanism [3] provides a suitable solution to better transmit the fusion of features in a global sequence. However, this causes a significant problem: the transformer has a large computing overhead, and the model is difficult to converge, which requires a large amount of computing resources. This complicates the deployment of mobile robots [6,7].

Therefore, we propose an effective grasp detection network based on improved a deformable convolution and spatial feature center mechanism (DCSFC-Grasp), which introduces deformable convolution for adaptive receptive field adjustment. Global long-range dependencies are captured using a lightweight multilayer perceptron (MLP) architecture in the spatial feature center (SFC), and a learnable feature center (LFC) mechanism is used to gather local regional features in the layers to capture partial geometric information and the entire visual appearance of the captured object. Unlike previous studies on robot grasping, in which the required grasping was predicted as a rectangle calculated by selecting the best option from multiple grasping probabilities, our network generates three images of the grasping quality, angle, and width, from which we can directly deduce the grasping rectangle of multiple grasping objects simultaneously, thus reducing the overall reasoning time. We provide detailed experimental evidence that our grasp detection network performs excellently on popular grasp datasets such as the Cornell and Jacquard datasets. The experimental results show that the SFC mechanism is indispensable in generating appropriate grasping rectangles by learning the local and global features of different parts of each object. Moreover, our grasp detection model is effective on actual robot systems and exhibits a good generalization ability for unknown objects.

The key contributions of our study can be summarized as follows:We propose a grasping system that can directly deduce the grasping rectangles of multiple grasping objects by generating images of the grasping quality, angle, and width;Our model introduces improved deformable convolution in the feature extraction module to adjust the adaptive receptive field, then uses an SFC layer to capture the global remote dependence through a lightweight MLP architecture. Compared with transformer encoders based on a multihead attention mechanism, the lightweight MLP architecture is simpler, lighter, and more computationally efficient. Additionally, to preserve the local corner region, we propose an LFC mechanism to gather local regional features in the layer. A lightweight CARAFE operator is used to complete the upsampling process. Compared with transpose convolution, the CARAFE operator has lower computational complexity and achieves better performance;We evaluated our model on publicly available grasp datasets and achieved the highest accuracies of 99.3% and 96.1% for the Cornell and Jacquard grasp datasets, respectively;We deployed the proposed model on an actual robot arm and conducted real-time grasping, which proved the feasibility of the model.

## 2. Related Studies

### 2.1. Grasp Detection with Deep Learning

Grasp detection refers to generation of the posture of the gripper of a manipulator to accomplish a grasping task by combining visual information with relevant algorithms [8]. It is primarily divided into two methods: analytical and empirical techniques. The analytical method refers to the analysis of the geometric structure of three-dimensional (3D) data according to various parameters of the manipulator and the generation of an appropriate grasping pose through the constraint of the design of force closure conditions [9]; however, most research in this area is based on an ideal model. On one hand, the variability of the actual scene, the randomness of the object placement, and the noise of the image sensor increase the complexity of the calculation; on the other hand, the accuracy of the analysis method cannot be guaranteed. Different empirical methods use prior information to detect the grasping pose and determine its rationality. Beginning with the features of objects, similarity is used for classification and pose estimation to achieve grasping. Recently, the development of deep-learning algorithms has provided strong technical support for empirical methods, and grasp detection using empirical methods has become the mainstream research method [10]. In addition, many factors must be considered in the grasp detection task, such as the physical characteristics of the object and types of grasping ends, most of which are suction cups and parallel grippers. Grasp detection methods applied to parallel grippers have been widely studied because of their dexterity.

Two-dimensional (2D) plane grasping methods use RGB or RGB-D image information and are limited to vertical grasping. For example, Jiang et al. [11] proposed a five-dimensional representation of plane-grasping rectangles and an algorithm to predict the grasping posture of a given target from an image. In [12], the authors proved that the five-dimensional grasping of 2D images can be manifested in 3D space, and they used deep learning algorithms to implement plane grasp detection on the Cornell dataset. In [2], a grasp detection network based on Alexnet [13] was proposed. Although the accuracy was significantly improved, the results of this method are more inclined to detect the central part because the direct regression model is often affected by the average annotation information. Kumra et al. [1] used a deeper ResNet50 [14] network as a feature extraction layer and conducted experiments based on a multimodal input, achieving accuracies of 88.8% and 89.2%. In [15], accuracy was further improved to 93.2% by correlating each grid cell with a default contrasted rectangle, such as in Faster-RCNN [16]. Chu et al. [17] adopted the default reference rectangle strategy and used a classification method to overcome the difficult problem of angle regression; the accuracy of the model reached 96.0%. Zhou et al. [18] designed a full convolution grasp detection network based on ResNet101 [14] in combination with the above concepts and achieved an accuracy of 97.7% on the Cornell grasp dataset; however, the efficiency was very poor. A common problem with these methods is that the network layer is deep, and the algorithm efficiency is low. Instead of using a deep CNN model to improve accuracy, Asif et al. [19] proposed an efficient convolution network architecture that uses fewer computing resources to achieve grasping prediction. The method achieved an accuracy of 90.2% on the Cornell grasp dataset at a speed of 24 ms per frame. In [20], a generative residual convolutional neural network (CNN) model, GR-ConvNet, which can generate grasping rectangles at a speed of 20 ms per frame, was proposed. Based on an RGB-D input, the accuracy of this method reached 97.7%; however, based on an RGB input, the accuracy was only 88.7%.

Recently, Wang et al. [21] proposed an architecture based on a transformer, TF-Grasp, which can obtain local information (i.e., the outline of an object) simultaneously. Furthermore, long-distance connections were modeled, and higher accuracies of 97.99% and 94.6% were obtained for the Cornell and Jacquard grasp datasets, respectively. However, some problems remain, such as high computational costs and difficulty in deploying mobile robots.

In addition, in recent years, some researchers have used object segmentation and pose estimation methods to achieve robotic grasp. In [22], the authors proposed a structural hybrid technique for a 3D recognition system based on the BiLuNetICP pipeline. With object detection and segmentation based on BiLuNet and accurate 3D measurement from the depth camera, their approach is able to provide robust results in 6D pose estimation. Tian et al. [23] proposed a pose estimation method based on crucial point detection, CenterNet-SPF, which divides the grasping angle into 18 feature point types and introduces subpixel convolution instead of transposed convolution to perform dual-channel feature fusion for high- and low-level features. The prediction accuracy reached 98.31% on the Cornell grasp dataset.

### 2.2. Deformable Convolutional Networks

The convolution kernel is an important component of a CNN. An increasing number of researchers have investigated methods of improving the convolution kernel to enhance the invariant feature extraction ability of CNNs. Dai et al. [24] proposed deformable convolutional networks (DCNs), which significantly improve the ability of CNNs to extract invariant features. A deformable convolution operation is an extension of the standard convolution operation. For plane figures, deformable convolution adds the learned offset to the mesh sampling location in a typical convolution such that the sampling location of the standard convolution is deformed and more concentrated in the region of interest (ROI). Thus, the sampling point of the convolution kernel changes its position based on the shape of the input image, forming an adaptive change ability and improved geometric invariance.

DCNs and CNNs have the same inputs and outputs. However, a DCN has a new convolutional layer and a fully connected layer for learning offsets. However, owing to the finite adaptive learning ability of the convolution kernel [25], DCNsv1 still contains content unrelated to the image content. The advantage of DCNs over CNNs lies in their ability to adapt to geometric changes in objects. The feature sampling area is closer to the shape of the objects, but the sampling area may significantly exceed the ROI, resulting in the features being affected by irrelevant image content. Zhu et al. [26] proposed DCNsv2 to improve DCNsv1 by integrating a deformable convolution more comprehensively within the network and introducing a modulation mechanism to expand the scope of deformable modeling. This comprehensively enhances the geometric modeling capability of deformable convolution and improves the ability of the network model to focus on key areas of an image. The deforming convolution module in DCNsv2 increases the adjustment mechanism, which can learn both the offset of the sampling points and the weight of each sampling point.

### 2.3. MLP in Computer Vision

To mitigate the limitations of complex transformer models [27,28,29,30], recent studies [31,32,33,34] have shown that good performance can still be achieved by replacing attention-based modules in transformer models with an MLP. This is because both the MLP (e.g., two fully connected layer networks) and the attention mechanism are information processing units. On one hand, the introduction of MLP-Mixer [31] mitigates the change in the data layout. On the other hand, MLP-Mixer can better establish the long dependency/global and spatial relationships of features through the interaction between spatial and channel feature information. Although MLP-style models excel in computer vision assignments, they still have limitations in capturing fine-grained feature manifestations and achieving a better detection effect [35].

Nonetheless, MLPs play an increasingly important role in the computer vision space and have the advantage of a plainer network structure compared with the transformer. In our study, we also used an MLP to capture the global context information and remote dependencies of input images. Our aim was to capture the centrality of information using the proposed SFC mechanism.

## 3. Improved Deformable Convolution and Spatial Feature Center Mechanism

### 3.1. Problem Statement

In this paper, we use a grasping system that can directly deduce the best grasping posture of unknown grasping objects by generating images of the grasping quality, angle, and width and executing them on a robot. Unlike the five-dimensional grasping representation proposed in [1,2,9], in this paper, we adopt an improved grasping representation similar to that developed in [36]. We represent the grasping posture in the robot frame as
(1)$$\begin{array}{c} G=(p,\theta_{r},\omega_{r},q), \end{array}$$
where p=(x,y,z) is the central location of the end of the robotic-arm two-finger gripper in the Cartesian coordinate system, $\theta_{r}$ is the azimuth angle of the gripper around the *z* axis, and $\omega_{r}$ is the width of the gap required by the two-finger gripper. In addition, the grasp quality score (*q*) is used to generate the probability of a successful grasp, and the optimal grasp configuration can be obtained using $G^{*}=\arg\max_{q}G$.

The grasping posture is detected using the n-channel image ($I=R^{n\times h\times w}$). Therefore, p=(x,y,z) can be simplified to 2D coordinates ($p_{i}=\left(x,y\right)$), which can be further defined as
(2)$$\begin{array}{c} G_{i}=\left(p_{i},\theta_{i},\omega_{i},q\right), \end{array}$$
where $p_{i}=\left(x,y\right)$ is the central point of the grasping rectangle corresponding to the image coordinates, $\theta_{i}$ is the grasping angle at position *i*, $\omega_{i}$ is the grasping width, and *q* is the grasping quality score.

$\theta_{i}$ represents the angle rotation required to grasp the target in each pixel within a range of $\left[\frac{-\pi }{2},\frac{\pi }{2}\right]$. To maintain the one-to-one mapping of the rotation angle ($\theta_{i}$) in the range of $\left[\frac{-\pi }{2},\frac{\pi }{2}\right]$, we decode the rotation angle learning into two components: sin(2θ) and cos(2θ). Thus, the final rotation angle is inversely derived from arctan(sin2θ/cos2θ)/2. $\omega_{i}$ is the width of the opening required for clamping, which is expressed as a measure of depth and is within the range of $\left[0,W_{\max}\right]$ pixels. $W_{\max}$ is the maximum width of the two-finger gripper. *q* measures the grasping success rate of each pixel in the image, and a value closer to 1 indicates a higher grasping success rate.

All the grasping sets can be expressed as follows:(3)$$\begin{array}{c} G=\left(\Theta ,W,Q\right)\in R^{3\times h\times w}, \end{array}$$
where Θ, *W*, and *Q* represent the predictive heat maps of the grasped objects, which are the grasping angle, width, and quality scores, respectively, and are calculated for each pixel of the image using Equation (Equation 2).

Using the grasping posture generated in the image space, the robot can perform grasping by transforming the image coordinates into the robot base coordinate system:(4)$$\begin{array}{c} G_{r}=T_{rc}\left(T_{ci}\left(G_{i}\right)\right), \end{array}$$
where $T_{ci}$ is the transformation of 2D space into 3D space of the camera using the inherent parameters of the depth camera, and $T_{rc}$ is the conversion of the camera space into robot space using the camera posture calibration value.

### 3.2. Network Architecture

The overall network architecture of DCSFC-Grasp is shown in Figure 1. It comprises an improved deformable convolution-based feature extraction module, SFC, several residual blocks, and a CARAFE upsampling operator. Grasp detection networks are suitable for any type of input mode. The pixel-by-pixel grasp position is generated by the grasp quality, angle, and width heat map. First, an input with a size of *n* × 224 × 224 (*n* = 1, 3, 4) is input into the improved deformable convolution-based downsampling module for feature extraction; then, the feature map is sent to the SFC layer to capture the global long-range dependence through a lightweight MLP architecture. Additionally, to retain the local feature information, we propose an LFC mechanism to aggregate the local features within the layer; then, the output of the SFC is transferred to five residual blocks. Finally, to easily interpret and reserve the spatial features after the convolution operation, we upsample the feature map using the CARAFE operator. Thus, we obtain the same image size on the output side as that on the input side.

#### 3.2.1. Improved Deformable Convolution-Based Feature Extraction Module

In a recent grasp detection work, GR-ConvNet [20], based on a CNN, three convolutional layers were used for downsampling to extract features. Although the features extracted using this method have strong semantics, some details are lost. For grasp detection, knowing the details of an unknown object is very important in order to predict a good grasping posture. Therefore, in this paper, a DCN is introduced in the feature extraction part for adaptive receptive field adjustment to obtain detailed feature information. Inspired by the research reported in [37], a shared projection weight is added based on DCNv2. Similar to conventional convolution, different sampling points in DCNv2 have independent projective weights; therefore, the size of their parameters is linearly related to the total number of sampling points. To reduce the complexity of the parameters and memory, we refer to the concept of separable convolution and use position-independent weights instead of grouping weights to share projective weights among different sampling points to preserve all sampling position dependencies.

The details of the feature extraction module are shown in Figure 2. To obtain hierarchical feature maps, we use a convolutional layer behind the deformable convolutional layer to downsample the feature maps twice. The core operator of the deformable convolutional layer is the improved DCN. The sampling effect and modulation scale are predicted within the DCN by running the input feature (*x*) through a separable convolution (3 × 3 deep convolution followed by a linear projection). A simple method of bridging the gap between multihead self-attention mechanisms (MHSAs) in the convolution and transformer is to introduce long-distance dependence and self-adaption spatial polymerization into the standard convolution. Given the input $x\in R^{C\times H\times W}$ and present point $p_{0}$, the DCN can be expressed as
(5)$$\begin{array}{c} y\left(p_{0}\right)=\sum_{k=1}^{K}wm_{k}x\left(p_{0}+p_{k}+\Delta p_{k}\right), \end{array}$$
where *K* is the total number of sampling points, representing a single sampling point; $w\in R^{C\times C}$ represents the position-independent projection weight, which is shared among different sampling points, and all sampling position dependencies are preserved; $m_{k}\in R$ expresses the modulation scalar of the *k*-th sampling point; $p_{k}$ represents the *k*-th position of predefined grid sampling {(−1,−1),(−1,0),…,(0,+1),…,(+1,+1)} in regular convolution; and $\Delta p_{k}$ is the offset of the *k*-th mesh sampling position. Equation (Equation 5) indicates that for long-range dependence, the sampling shift ($\Delta p_{k}$) can flexibly connect with local feature information; therefore, we can conclude that the DCN and MHSA have similar favorable properties.

#### 3.2.2. Spatial Feature Center Module

Inspired by EVCBlock for object detection [38], as shown in Figure 3, the SFC layer proposed in this paper is primarily composed of two parallel-connected blocks, in which a lightweight MLP can capture the global remote dependencies of the top-level feature (*X*). To preserve the local corner area information, we use an LFC mechanism to integrate local regional features within layers. The resulting feature maps of the two parallel-connected blocks are joined together along the channel dimension, which can be formulated as
(6)$$\begin{array}{c} X_{out}=cat\left(MLP\left(X_{in}\right);LFC\left(X_{in}\right)\right), \end{array}$$
where cat(·) represents the feature-map cascade; $MLP(X_{in})$ and $LFC(X_{in})$ represent the output features of the lightweight MLP and LFC mechanism, respectively; and $X_{in}$ is the output of the stem block, which is used for feature smoothing rather than immediate implementation in the original feature map, as in [39]. It consists of a convolutional layer, batch normalization (BN) layer, and rectified linear unit (ReLU) layer.
(7)$$\begin{array}{c} X_{in}=\sigma \left(BN\left(Conv_{7\times 7}\left(X\right)\right)\right) \end{array}$$
where $Conv_{7\times 7}\left(\cdot \right)$ is a 7 × 7 convolutional layer, BN(·) represents the BN layer, and σ(·) represents the ReLU layer.

The lightweight MLP primarily comprises two residual modules: the module based on deep convolution [40] and the block based on the channel MLP. The two modules are connected in series. Both blocks are followed by channel scaling [33] and drop-path [41] operations. Specifically, for deep convolution-based modules, the features ($X_{in}$) output from stem blocks are first fed into a deep convolutional layer. Compared with traditional spatial convolution, deep convolution can improve feature representation and reduce computational costs. Subsequently, channel scaling and the drop path are implemented. Thereafter, residual joining of $X_{in}$ is implemented. This process can be formulated as
(8)$$\begin{array}{c} X_{d}=DConv\left(GN\left(X_{in}\right)\right)+X_{in}, \end{array}$$
where GN(·) denotes group normalization, and DConv(·) denotes deep convolution.

For the module based on the channel MLP, the features that are output from the module based on deep convolution are first transmitted to GN(·), and the channel MLP [31] is implemented on these features. Compared with spatial MLP, the channel MLP can effectively diminish the computational complexity and satisfy the demands of common visual assignments, after which the channel scaling, drop path, and residual connection to $X_{d}$ are achieved.
(9)$$\begin{array}{c} MLP\left(X_{in}\right)=CMLP\left(GN\left(X_{d}\right)\right)+X_{d}, \end{array}$$
where CMLP(·) is the channel MLP. We omit the channel scaling and drop path from Equations (Equation 8) and (Equation 9) for demonstration.

The LFC is an encoder with an intrinsic dictionary with two components: (1) an intrinsic code book ($B=\{b_{1},b_{2},\ldots ,b_{K}\}$) and (2) a set of scaling factors ($S=\{s_{1},s_{2},\ldots ,s_{K}\}$) of the feature center that can be learned. Specifically, $X_{in}$ is first encoded using a combination of a set of convolutional layers (comprising 1 × 1, 3 × 3, and 1 × 1 convolutional layers). Subsequently, the encoded features are processed using CBR blocks, which consist of a 3 × 3 convolutional layer, BN layer, and ReLU layer. Through these steps, the code feature ($X_{in}$) is entered into the code book. Thus, we utilize a set of scaling factors (*S*) to consecutively map $x_{i}$ and $b_{k}$ to the corresponding location information. The information regarding the entire image of the *k*-th code word can be calculated as follows:(10)$$\begin{array}{c} e_{k}=\sum_{i=1}^{N}\frac{e^{-s_{k}{\left|\right|x_{i}-b_{k}\left|\right|}^{2}}}{\sum_{j=1}^{K}e^{-s_{k}{\left|\right|x_{i}-b_{k}\left|\right|}^{2}}}\left(x_{i}-b_{k}\right), \end{array}$$
where $x_{i}$ is the *i*-th pixel point, $b_{k}$ is the *k*-th learnable feature code word, and $s_{k}$ is the *k*-th scaling factor. $x_{i}-b_{k}$ provides information about the position of each pixel relative to the code word. *K* is the total number of feature centers. Subsequently, we utilize ϕ(·) to aggregate all $e_{k}$ values, which contain the BN layer with the ReLU and average layers. Based on this, the complete information about the entire image of *K* code words is computed as follows:(11)$$\begin{array}{c} e=\sum_{k=1}^{K}\phi (e_{k}). \end{array}$$

After the output of the code book is acquired, we further transmit the *e* value to the fully connected and convolutional layers to predict the features that emphasize the critical classes. We then multiply it by the channels between the output feature ($X_{in}$) from the stem block and the scale factor (δ(·)). This process is represented as follows:(12)$$\begin{array}{c} Z=X_{in}⊗\left(\delta \left(Conv_{1\times 1}\left(e\right)\right)\right), \end{array}$$
where ⊗ denotes multiplication by channels, and δ(·) is the sigmoid function. Eventually, we perform channel addition between the feature ($X_{in}$) and the local corner area feature (*Z*), which is represented as
(13)$$\begin{array}{c} LFC\left(X_{in}\right)=X_{in}⊕Z, \end{array}$$
where ⊕ denotes the addition of channels.

### 3.3. Loss Function

Our objective is to obtain the predicted grasping heat map (*G*) from a set of input images ($I=\{I_{1},I_{2},\ldots ,I_{k}\}$). For the dataset consisting of the grasped object and grasped label ($L=\{L_{1},L_{2},\ldots ,L_{k}\}$), we aim to minimize the diversity between *G* and *L*. We analyzed the performance of various loss functions of the network and observed that after running several tests, the smooth L1 loss had the best performance in handling the explosion gradient. Therefore, the loss of the entire loss function can be represented as
(14)$$\begin{array}{c} \begin{array}{c} Loss\left(G,L\right)=\sum_{i}^{K}\sum_{m}Loss_{smothL1}\left(G_{i}^{m}-L_{i}^{m}\right)\\ m\in \{\Theta ,W,Q\}. \end{array} \end{array}$$
with $Loss_{smothL1}$:(15)$$\begin{array}{c} Loss_{smothL1}=\left\{\begin{array}{c} 0.5{\left(G_{i}^{m}-L_{i}^{m}\right)}^{2},\mathrm{if}\left|G_{i}^{m}-L_{i}^{m}\right|<1\\ \left|G_{i}^{m}-L_{i}^{m}\right|-0.5/\sigma^{2},\mathrm{otherwise}, \end{array}\right. \end{array}$$
where *K* is the total number of sampling points, and σ is the hyperparameter operating in the smoothing region.

## 4. Experimental Validation

We verified the performance of DCSFC-Grasp through numerous experiments. We compared current methods using two popular grasp detection datasets, studied the performance of the proposed module, and evaluated its effectiveness on an actual 6-DoF Realman RM65 manipulator.

### 4.1. Datasets and Experimental Setup

(1) Datasets: The number of publicly available grasp datasets is limited. We used the Cornell [10] and Jacquard [42] grasp datasets to train and validate our method. The Cornell dataset comprises 885 RGB-D images of 224 objects. The Jacquard dataset was generated in a simulator using a CAD model. It contains more than 50,000 images of 11,000 object classes and more than 1 million annotated grasp labels.

(2) Evaluation Metric: To fairly compare the results with those of other methods, we utilized the Jaccard index and azimuth threshold as evaluation indicators. A grasping configuration is deemed effective when it satisfies the following two circumstances:The intersection-over-union (IoU) score between the generated predictive grasping rectangle *G* and the ground truth grasping rectangle ($G_{gt}$) is greater than 0.25, that is,
(16)$$\begin{array}{c} IoU=\frac{\left|G\cap G_{gt}\right|}{\left|G\cup G_{gt}\right|}>0.25; \end{array}$$

The offset ($\Theta_{\Delta }$) between the predicted azimuth of the grasped rectangle (Θ) and the ground truth azimuth of the grasped rectangle ($\Theta_{gt}$) is less than $30^{\circ }$, that is,


(17)
$$\begin{array}{c} \Theta_{\Delta }=\left|\Theta -\Theta_{gt}\right|<30^{\circ }. \end{array}$$


(3) Network Training: The proposed DCSFC-Grasp was implemented using PyTorch 1.10.1 and CUDA 10.1, and the overall model was used in Ubuntu 18.04. We followed a common strategy to train DCSFC-Grasp on an NVIDIA A100-PCIE 40-GB GPU with a batch size of 36. AdamW was applied as the optimizer, and the learning rate was set to decrease as the training progressed, with an initial value of 5 × 10^−4^. DCSFC-Grasp uses a 224 × 224 RGB-D image as its input, and the output is three heat maps of the same size as the input.

### 4.2. Experimental Results and Analysis

Owing to the relatively small size of the Cornell dataset, we followed the setup of previous studies [1,2,11,21] using quintupled cross validation. In addition, we considered run times to make a comprehensive comparison. For all comparison models, we used the accuracy informed in their primitive articles. We used image-wise split (IW) and object-wise split (OW) cross-validation settings to split the dataset [43]. IW is used to trial a model’s predictive capability when objects have different attitudes, and OW is used to trial a model’s generalization capability when running into various objects. The detection accuracy of the Cornell dataset is summarized in Table 1.

From the table, we observe that compared with the existing grasp detection methods, DCSFC-Grasp achieved the highest accuracies of 99.3% and 98.5% for different evaluation indicators divided by IW and OW on the Cornell dataset, respectively, and the average reasoning speed of each image was 22 ms, which fully guaranteed the real-time grasp of the real robot.

To evaluate the robustness of our approach further, we compared DCSFC-Grasp with GR-ConvNet [20] and TF-Grasp [21] under various evaluation indicators, including the Jaccard index and azimuth threshold (Table 2). Note that when the Jaccard index reached 0.4, DCSFC-Grasp still maintained an accuracy of 98.5%, which means that the accuracy of the grasping position is very demanding. From the comparison results, we observed that our model achieved the best results under all evaluation indicators divided by IW and OW. In addition, as the Jaccard index increased, the azimuth threshold decreased, and the success rate of GR-ConvNet and TF-Grasp decreased rapidly, with DCSFC-Grasp maintaining a high detection accuracy. The results demonstrate that DCSFC-Grasp has a stable grasp detection capability.

Figure 4 depicts the grasp detection results of unknown targets using DCSFC-Grasp after training on the Cornell dataset. The three columns on the left are RGB images, generated grasping rectangles, and depth images, whereas the three columns on the right show the grasp width, angle, and quality score heat maps.

We conducted a comparison experiment for [20,21,44] on the grasping prediction results generated for unknown objects, as shown in Figure 5. As shown in the figure, DCSFC-Grasp generated correct grasp prediction rectangles for all four unknown objects. In contrast, the other three models had unsuccessful grasping predictions. For example, as shown in Figure 5(2), the grasp detection results of the first three models could not be successfully grasped, and only the proposed method predicted the rectangles that could be successfully grasped. Our method can generate grasping positions for simply shaped objects, such as staplers and pliers, which have high-quality heat maps. For objects with complex structures, such as tapes and game pads, DCSFC-Grasp can correctly capture the most appropriate grasp location at a suitable grasp width and angle. These advantages may be attributed to the ability of DCSFC-Grasp to learn locally and globally valid features of the grasped target, which is vital for the grasping task. The experiments showed that our model is suitable and robust for targets with various shapes.

Table 3 shows the accuracy of DCSFC-Grasp compared with currently available models on the Jacquard dataset. Our method achieved an accuracy of 96.1% and was superior to other methods, which indicates that the deformable convolution-based feature extraction module and SFC module in the proposed DCSFC-Grasp improve the grasping performance.

### 4.3. Comparison of Multiobject Grasp

Although both the Cornell and Jacquard datasets are single-object grasping datasets, to verify the strong accuracy and robustness of DCSFC-Grasp for multiobject grasping prediction, we directly used the existing algorithms to conduct comparison experiments, in which all objects had not been seen before. The generated grasping posture and mass heat maps are shown in Figure 6. We observed that existing methods produced inaccurate grasp positions in some scenarios and that the shadow of the object negatively affected their generation of appropriate grasp rectangles. In addition, they generated inadequate grasp angles and widths, which can lead to poor robustness and grasp failures when used in actual robots. In contrast, from the mass heat map, our method distinguished the object from the background well, was not affected by shadows, better understood the grasping scene, and more accurately predicted the grasping success rate of different positions of the object.

The detection results show that DCSFC-Grasp improved the performance. We infer that previous models may not have fully considered grasping location information, resulting in poor spatial representation and the omission of some critical features. In contrast, our method, in the entire process of information transmission, adopts a deformable convolution-based feature extraction module and an SFC module. Thus, grasping models can establish accurate relationships between grasping and features such as the form, outline, and location of targets, which are essential for successful grasping.

### 4.4. Ablation Study

In this section, we consider a more in-depth analysis of the different modules in DCSFC-Grasp from the following two perspectives to understand their impact on the performance of the overall model.

#### 4.4.1. Improved DCN vs. Convolutional Layer

In recent grasp detection works, (GG-CNN [44] and GR-ConvNet [20]), three convolutional layers were used for downsampling to extract features. We conducted an ablation study based on the Cornell dataset. We replace three convolutional layers in GG-CNN and GR-ConvNet with the improved deformable convolution-based feature extraction module and verify the effectiveness of the improved DCN through comparative analysis of precision. Specific information regarding the results of the ablation study is presented in Table 4.

When the convolutional layer in the model is replaced by the improved DCN, the accuracy of the two models is improved to different degrees. For grasp detection, obtaining the details of an unknown object is very important for prediction of a good grasping posture. Compared to using convolutional layers to extract features, the deformable convolution-based feature extraction module adaptively adjusts the receptive field to obtain detailed feature information.

#### 4.4.2. Effectiveness of the SFC Module

The SFC module is primarily composed of two parallel connection blocks, namely a lightweight MLP and learnable feature center (LFC) blocks. The resulting feature maps of the two parallel-connected blocks are joined together along the channel dimension, with the stem block in front of them. In this section, we perform ablation studies based on the Cornell dataset to understand the role of each sub-block in the SFC module.

The details of experimental results are shown in Table 5, showing that the three sub-blocks in the SFC module are essential to improve the performance of DCSFC-Grasp. The proposed SFC can focus on the shape and position information about the feasible grasp area and help our model locate a position that is easy to grasp.

### 4.5. Grasping in Real-World Scenarios

We conducted physical experiments using the 6-DoF Realman RM65 robotic arm and a Microsoft Azure Kinect DK RGB-D camera mounted with in the eye-out hand manner to maintain complete field-of-view coverage of the grasping object. We employ the well-trained grasp model in an independent thread, which communicates with the camera and other robot threads through the ROS topic mechanism to subscribe images and publish grasp poses. The experimental setup is shown in Figure 7. In each grasp task, DCSFC-Grasp accepted a visual signal from the camera and output the optimal grasp rectangle. The coordinates of the grasp rectangle in the camera coordinate system were converted to those under the arm base. Thereafter, the robot arm end gripper approximated the optimum target grasp position according to the motion trajectory planned by the manipulator motion-path-planning method, and the target was grasped by closing the two-finger gripper. Therefore, our grasp system can be conveniently applied to other hardware platforms.

We conducted extensive robot grasping experiments in real-world scenarios with other methods. The success rates are presented in Table 6. We conducted 400 grasping tasks on a real robot arm, of which 382 were successful, corresponding to a success rate of 95.5%. Compared to other methods, the results demonstrate that our DCSFC-Grasp grasp detection system performs well on practical robotic arms.

## 5. Conclusions

This paper proposes a high-performance object grasp detector, DCSFC-Grasp, for prediction of optimal grasp positions for unknown objects. Owing to the modules developed, DCSFC-Grasp can focus on capturing location features and ensuring the complete and effective dissemination of information. Thus, a robot can correctly capture the optimum grasping location and generate appropriate grasping widths and angles for different targets. We evaluated DCSFC-Grasp on a public grasping dataset, and its performance was found to be better than that of state-of-the-art methods. In addition, we conducted several robot grasping tasks in various scenarios, further demonstrating that DCSFC-Grasp can generate and execute precise grasping. In conclusion, compared with existing models, our model can learn to capture relevant features more efficiently and achieve higher accuracy. However, it is currently only suitable for two-finger fixtures. In future research, we will apply DCSFC-Grasp to five-finger dexterous hands to complete more difficult grasping tasks.

## Figures and Tables

**Figure 1 biomimetics-08-00403-f001:**
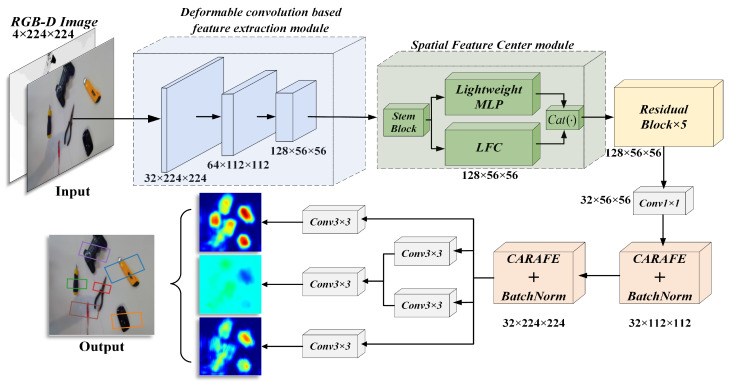
Overall architecture of DCSFC-Grasp. The improved deformable convolution-based feature extraction module introduces an improved deformable convolution to adaptively adjust the receptive field to obtain detailed feature information. The spatial feature center (SFC) module uses a lightweight MLP to capture the global long-range dependence of the top-level feature information, while a learnable feature center (LFC) mechanism is used to aggregate local regional features within the layer to preserve local corner regions. Finally, the CARAFE operator is applied to upsample the feature map and generate a heat map with the same resolution as the input.

**Figure 2 biomimetics-08-00403-f002:**
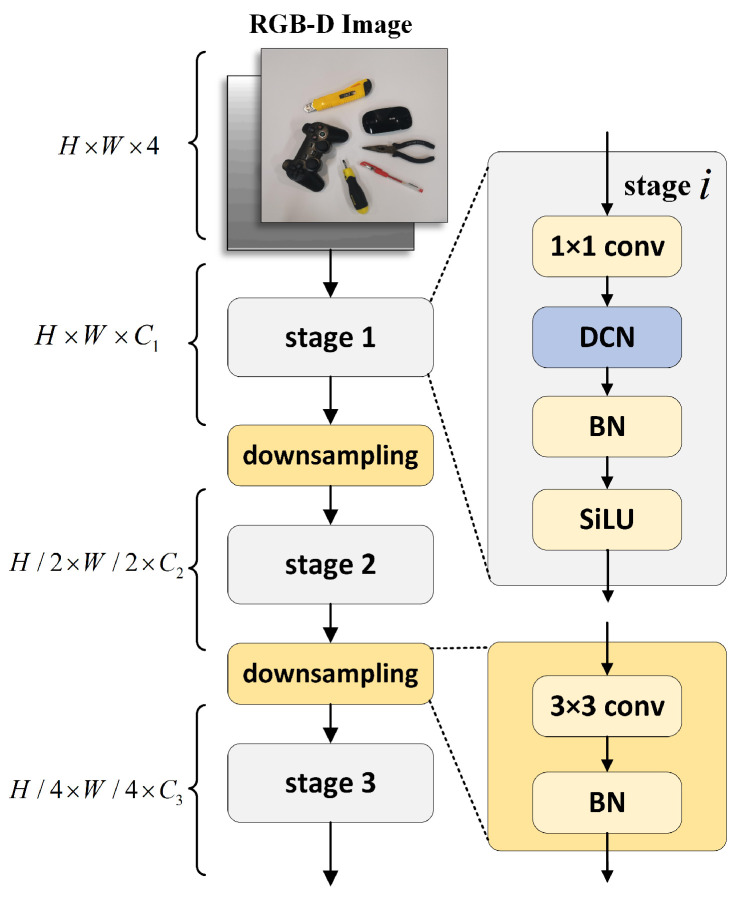
Structure of deformable convolution-based feature extraction module. To obtain the layered feature map, we use a convolutional layer behind the deformable convolutional layer to downsample the feature map twice. The core operator of the deformable convolutional layer is the improved DCN.

**Figure 3 biomimetics-08-00403-f003:**
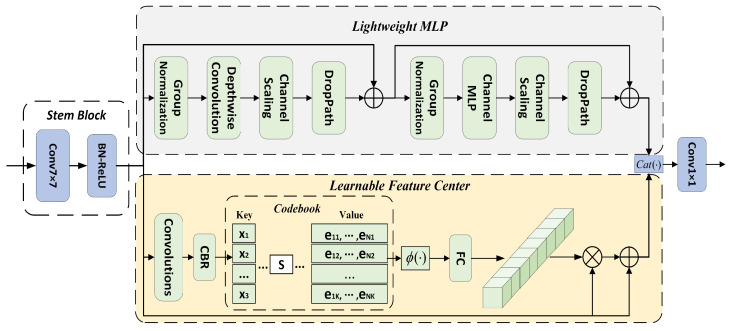
Structure of the SFC, which is primarily composed of two parallel connected blocks, in which the lightweight MLP can capture the global remote dependencies of the top-level feature. An LFC is used to integrate the local regional features within layers to preserve local corner area information. The resulting feature maps of the two parallel-connected blocks are joined together along the channel dimension. The stem block at the front is used for feature smoothing.

**Figure 4 biomimetics-08-00403-f004:**
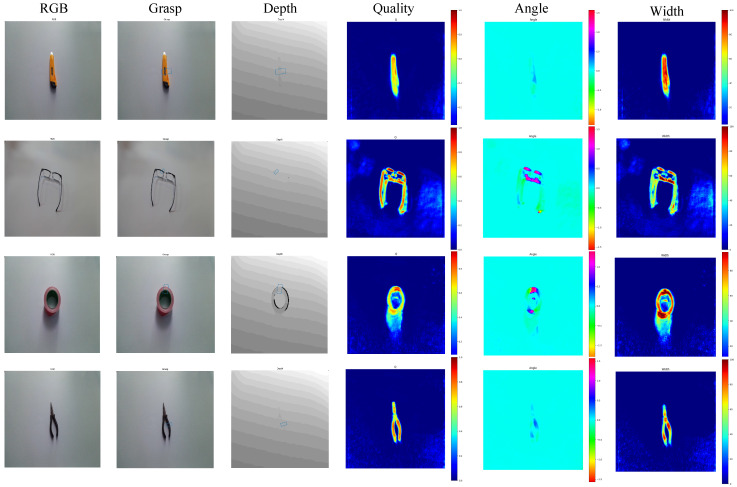
Grasp detection results of unknown targets using DCSFC-Grasp after training on the Cornell dataset.

**Figure 5 biomimetics-08-00403-f005:**
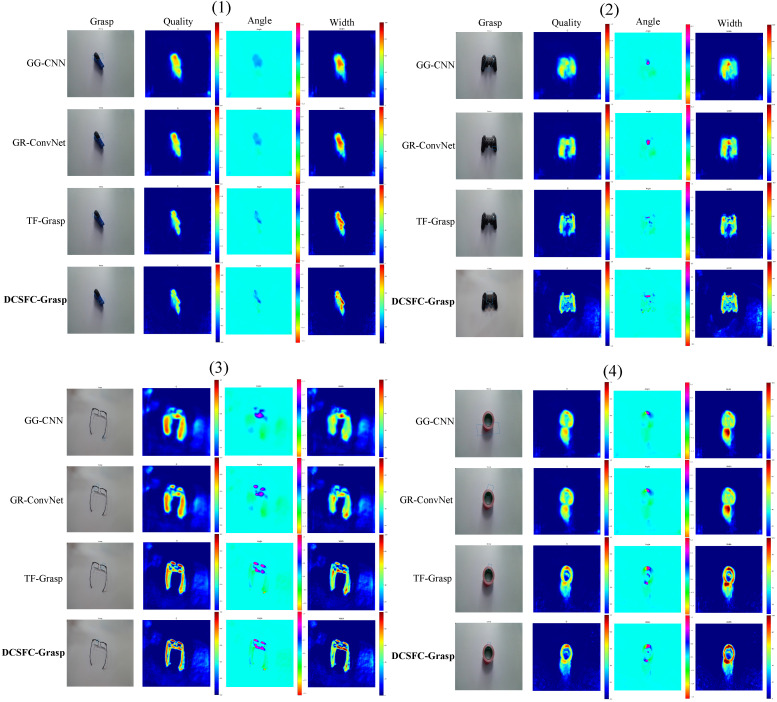
Results of the comparison of grasp predictions for unknown targets on the Cornell dataset.

**Figure 6 biomimetics-08-00403-f006:**
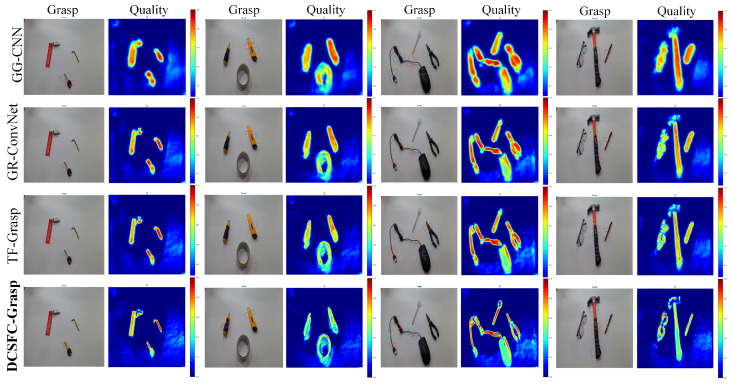
Comparison of grasp detection results for unknown multiple objects.

**Figure 7 biomimetics-08-00403-f007:**
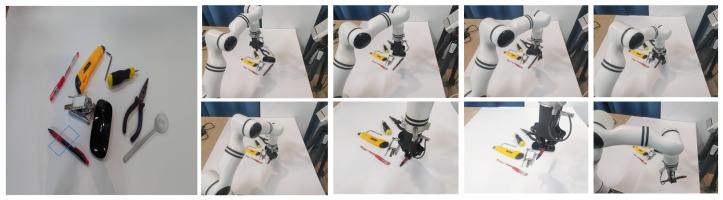
Screenshots of grasping in real-world scenarios.

**Table 1 biomimetics-08-00403-t001:** Comparison of results for the Cornell dataset.

Method	Author	Accuracy (%)		Time (ms)
		IW	OW	
SAE [12]	Lenz	73.9	75.6	1350
AlexNet, MultiGrasp [2]	Redmon	88.0	87.1	76
GG-CNN [44]	Morrison	73.0	69.0	19
GRPN [45]	Karaoguz	88.7	-	200
ResNet-50x2 [1]	Kumra	89.2	88.9	103
GR-ConvNet-RGB-D [20]	Sulabh	97.7	96.6	20
E2E-net-RGB [46]	Ainetter	98.2	-	63
TF-Grasp [21]	Wang	97.99	96.7	42
SE-ResUNet [47]	Yu	98.2	97.1	25
CenterNet-SPF [23]	Tian	98.31	97.6	24
DCSFC-Grasp	Ours	99.3	98.5	22

**Table 2 biomimetics-08-00403-t002:** Comparison of accuracy at different Jaccard indices and azimuth thresholds using the Cornell dataset.

Method	Splitting	Jaccard Index					Azimuth Threshold				
		20%	25%	30%	35%	40%	10°	15°	20°	25°	30°
GR-ConvNet [20]	IW (%)	98.1	97.7	96.8	94.1	88.7	86.4	94.4	97.2	97.7	97.7
TF-Grasp [21]		98.4	97.99	97.2	94.9	90.3	89.21	95.44	97.52	97.98	97.99
DCSFC-Grasp		99.5	99.3	99.2	99.0	98.5	93.6	94.8	98.3	99.2	99.3
GR-ConvNet [20]	OW (%)	97.1	96.6	93.2	90.5	84.8	84.7	90.9	95.1	95.8	96.6
TF-Grasp [21]		97.4	96.7	93.9	91.7	85.2	85.2	91.6	95.4	96.0	96.7
DCSFC-Grasp		98.6	98.5	98.3	98.1	97.4	93.1	93.9	97.4	98.5	98.5

**Table 3 biomimetics-08-00403-t003:** Results of the comparison for the Jacquard dataset.

Method	Author	Year	Accuracy (%)
Jacquard [42]	Depierre	2018	74.2
GG-CNN [44]	Morrison	2018	84.0
FGGN,ResNet-101 [18]	Zhou	2018	91.8
GR-ConvNet [20]	Sulabh	2020	94.6
RSEN [48]	Cao	2021	94.8
TF-Grasp [21]	Wang	2022	94.6
SE-ResUNet [43]	Yu	2022	95.7
CenterNet-SPF [23]	Tian	2022	95.5
DCSFC-Grasp	Ours	2023	96.1

**Table 4 biomimetics-08-00403-t004:** Ablation study of improved DCN on the Cornell dataset.

Network	Accuracy (%)	
	IW	OW
GG-CNN (convolutional layer)	73.0%	69.0%
GG-CNN (improved DCN)	79.6%	77.2%
GR-ConvNet (convolutional layer)	97.7%	96.6%
GR-ConvNet (improved DCN)	98.1%	97.2%

**Table 5 biomimetics-08-00403-t005:** Ablation study of SFC in the proposed modules.

Lightweight MLP		✓			✓		✓	✓
Learnable Feature Center Block			✓		✓	✓		✓
Stem Block				✓		✓	✓	✓
Accuracy (IW)	97.8%	98.0%	98.7%	97.9%	99.1%	98.8%	98.2%	99.3%

**Table 6 biomimetics-08-00403-t006:** Physical grasping results in real-world scenarios.

Method	Physical Grasp	Success Rate (%)
GG-CNN [44]	167/200	83.5
GR-ConvNet [20]	172/200	86.0
TF-Grasp [21]	152/165	92.1
SE-ResUNet [43]	369/400	92.3
Ours	382/400	95.5

## Data Availability

Datasets used in this paper are available from: http://pr.cs.cornell.edu/grasping/rect_data/data.php (accessed on 10 August 2021) and https://jacquard.liris.cnrs.fr/ (accessed on 15 February 2022).

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
