# Peer review of "Robotic Grasp Detection Network Based on Improved Deformable Convolution and Spatial Feature Center Mechanism"

_biomimetics, 2023, doi:10.3390/biomimetics8050403_

Round 1

Reviewer 1 Report

The author proposed a high-performance object grasp detector, DCSFC-Grasp, for predicting optimal grasp positions for unknown object. After training and developing the modules, it can focus on capturing location features and ensuring the complete and effective dissemination of information. Here are several issues in my opinion.

1.     Format and layout.

Good format and layout can make a manuscript easy to understand. Some parts and images in this manuscript are not intuitive and aesthetically pleasing.

·         There is a lot of blank space after line 208, page 5.

·         The image size in figure 1 is too large, which looks uncomfortable. As well as figure 5,6,7.

2.     Experiment

·         In line 298, page 12.” In contrast, the other three models had unsuccessful grasping predictions.” As shown in figure 5, the grasp predictions of TF-Grasp model are similar with the DCSFC-Grasp’s. so why the author say the grasp predictions of first three model are unsuccessful?

·         Could you define the ablation study?

·         The author conducted physical experiments using the 6-DoF Realman RM65 robotic arm and Microsoft Azure Kinect DK RGB-D camera. Could you show more details about the robotic arm and camera?

·         To make comparisons, the author shows many other methods in table 1 and 3. Did they do any physical experiments? If so, what about their success rate in real-world scenarios in comparison?

It looks good.

Reviewer 2 Report

This paper presents a robot grasp detection network by the use of deformable convolution and spatial feature center mechanism.The proposed DCSFC-Grasp aims to detection the grasping position of an object for robot manipulation. With the RGB-D image input, the network model is designed and implementation. The experiments are carried out with Cornell and Jacquard grasp datasets to illustrate the effectiveness of the approach. Although some good results are provided, it would be beneficial by comparing with object segmentation and pose estimation for robotic grasp, such as in "Van Tran, L. and Lin, H.Y., 2020. BiLuNetICP: a deep neural network for object semantic segmentation and 6D pose recognition. IEEE Sensors Journal, 21(10), pp.11748-11757." The results are currently compared using Cornell dataset for several different techniques. It is expected to adopt more recent works for performance evaluation. For the network architecture, the ablation study is relatively limited. More in-depth analysis on different modules in the model can provide better understanding of the proposed method. Finally, the impact of the paper and journal will be increased if making the code available publicly.

Moderate editing of English language required.

Reviewer 3 Report

This paper proposes one grasp detection network based on improved deformable convolution and spatial feature center mechanism for robot grasping, and carries out the experimental validation. The whole document is well structured. The following comments shall be revised:

(1)Please shorten the tittle of Sec. 3, and one schematic of grasping system is appreciated.

(2)Check the expression of “grasping” or “grasp”.

(3)Please update the tittle of Sec.4 as “Experimental validation”.

The quality of English language is very good.
